# Outcomes of posterior lamellar tarsal rotation vs bilamellar tarsal rotation for trachomatous trichiasis

John H. Kempen[1,2,3,4,5]*, Yineng Chen[6], Aida Abashawl[7], Ahlam Awad Mohammed[7], Sarity Dodson[8,9], Wondu Alemayehu[8], Alemu Gemechu[8], Aemero Abateneh Mengesha[7,10], Dereje Adugna Kumsa[11], Tony Succar[12], Kathleen McWilliams[6], Fangming Jin[6], Vatinee Y. Bunya[6], Maureen G. Maguire[6], Matthew J. Burton[13,14], Gui-shuang Ying[6,15], FLuorometholone as Adjunctive MEdical Therapy for Trachomatous Trichiasis Surgery (FLAME) Trial Research Group¶

1 Department of Ophthalmology, Schepens Eye Research Institute, Massachusetts Eye and Ear Infirmary, Boston, Massachusetts, United States of America, 2 Department of Ophthalmology, Harvard Medical School, Boston, Massachusetts, United States of America, 3 Sight for Souls, Bellevue, Washington, United States of America, 4 MCM Eye Unit; MyungSung Christian Medical Center (MCM) General Hospital and MyungSung Medical College, Addis Ababa, Ethiopia, 5 Department of Ophthalmology, Addis Ababa University School of Medicine, Addis Ababa, Ethiopia, 6 Scheie Eye Institute/Department of Ophthalmology, The Perelman School of Medicine, University of Pennsylvania, Philadelphia, Pennsylvania, United States of America, 7 Berhan Public Health and Eye Care Consultancy, Addis Ababa, Ethiopia, 8 Fred Hollows Foundation, Addis Ababa, Ethiopia, 9 Fred Hollows Foundation, Melbourne, Australia, 10 Department of Ophthalmology, Bahir Dar University College of Medicine and Health Sciences, Bahir Dar, Ethiopia, 11 Oromia Regional Health Bureau, Addis Ababa, Ethiopia, 12 Massachusetts Eye and Ear Infirmary, Boston, Massachusetts, United States of America, 13 International Center for Eye Health, London School of Hygiene & Tropical Medicine, London, United Kingdom, 14 National Institute for Health Research Biomedical Research Centre for Ophthalmology at Moorfields Eye Hospital NHS Foundation Trust and UCL Institute of Ophthalmology, London, United Kingdom, 15 Department of Biostatistics and Epidemiology, Center for Clinical Epidemiology and Biostatistics, The Perelman School of Medicine, The University of Pennsylvania, Philadelphia, Pennsylvania, United States of America

¶ Membership of the FLAME Trial Research Group is provided in Supporting Information file S1 Acknowledgments in S1 File.
* John_Kempen@meei.harvard.edu

## Abstract

### Background

Trachomatous trichiasis (TT) surgery is a key strategy for avoiding blindness and visual impairment from trachoma. We compared alternative WHO-endorsed TT surgery techniques, hypothesizing that in a "real world" study posterior lamellar tarsal rotation (PLTR) would be associated with less postoperative TT (PTT) than bilamellar tarsal rotation (BLTR).

### Methods

In an ongoing TT control program in Jimma Zone, Ethiopia, TT surgeons used their preferred procedure (PLTR or BLTR) for upper eyelids with TT. Logistic regression—crude or adjusting for inter-eye correlation and relevant baseline factors (age, number

**Data availability statement:** All relevant data are in the manuscript and its supporting information files.

**Funding:** Primary funding from National Eye Institute/National Institutes of Health (Bethesda, Maryland, USA) grants UG1EY030420 (J.H.K.) and UG1EY030419 (G.S.Y.). Additional funding from the Massachusetts Eye and Ear Global Surgery Program (Boston, Massachusetts, USA), Sight for Souls (Bellevue, Washington, USA), and Research to Prevent Blindness (New York, New York, USA). The FLAME Trial is receiving donations of FML from Allergan (an Abbvie company, North Chicago, IL, USA). The sponsors did not play any role in the study design, data collection and analysis, decision to publish, or preparation of the manuscript except that the National Eye Institute selected the Data and Safety Monitoring Committee and sends Program Officers to participate in the study's Executive Committee.

**Competing interests:** I have read the journal's policy and the authors of this manuscript have the following competing interests: JHK owns Betaliq and Tarsier stock. Betaliq is developing an intraocular pressure-lowering eyedrop. Tarsier is developing an anti-inflammatory eyedrop. Neither is marketing their products at this point. VYB is a member of the Advisory Board for Kowa.

of trichiatic lashes, epilation, entropion severity, and upper palpebral conjunctival scarring severity)—was used to compare the one-year cumulative incidence of PTT (any upper eyelid lash touching the globe, evidence of epilation and/or repeat TT surgery).

## Findings

Most baseline TT severity markers were worse in the PLTR (855 eyes) than the BLTR (678 eyes) group and PLTR surgeons were less experienced than BLTR surgeons. Nevertheless, one-year cumulative PTT incidences were 8.2% (PLTR) and 21.4% (BLTR; adjusted odds ratio = 0.27, 95% confidence interval: 0.19-0.39). Prospectively ascertained postoperative adverse TT surgery outcomes were similar between groups by six months and 12 months postoperatively.

## Interpretation

When surgeons applied their preferred surgical technique, PTT occurred less than half as often with PLTR than BLTR. These real-world data confirm a prior trial's primary result, suggesting that using PLTR instead of BLTR reduces PTT incidence to a clinically important degree without increasing adverse outcomes. Another recent trial suggests continued BLTR is appropriate for surgeons already trained in that technique.

## Trial registration

www.clinicaltrials.gov, NCT04149210.

## Author summary

Trachoma is the leading infectious cause of blindness world-wide, affecting the poor. Chronic inflammation from trachoma can lead to scarring of the covering of the white part of the eye and inner surface of the eyelids. Scarring then may lead to in-turning of the eyelashes which scratch the eye surface ("trichiasis"). This scratching is the primary pathway leading to blindness. Trachoma elimination programs include eyelid surgery to out-turn the upper eyelashes back into the right position. Unfortunately, sometimes the eyelash scratching returns after surgery. We compared the two leading ways of doing this corrective surgery. In the first year after surgery, only 8.2% of 855 eyes having the surgery with cutting only on the inner layers of the upper eyelids ("posterior lamellar tarsal rotation") had recurrence of eyelash scratching while 21.4% of 678 eyes having surgery cutting all the way through the upper eyelid ("bilamellar tarsal rotation") had recurrence. Other problems after surgery were similarly uncommon in the two groups at 12 months after surgery. Combined with evidence from previous research which found similar results, our study suggests that patients have better

outcomes from the surgery on the inner layers of the eyelid unless the surgeon already is familiar with the technique cutting all the way through the lid.

## Introduction

Trachoma, a neglected tropical disease, is the leading infectious cause of blindness—mediated primarily by trachomatous trichiasis. [1–4]. Trachomatous trichiasis (TT) is a late-stage complication of chronic trachoma, which occurs in a minority of the trachoma-affected public, causing vision loss by damaging the cornea. It also causes severe and disabling pain and/or foreign body sensation [4,5].

World Health Organization (WHO)-endorsed programmatic efforts to avoid blindness and visual impairment from trachoma use a four-pronged program under the acronym SAFE (Surgery, Antibiotics, Face-washing and Environmental changes) [5]. Eyelid rotation surgery to alleviate TT and community-wide antibiotic distribution to reduce the number of persons with active trachoma are conducted primarily through vertical programs, whereas face-washing and environmental changes usually are implemented as two of many goals carried out by the Water, Sanitation and Hygiene (WASH) sector and general public health efforts.

Trachomatous trichiasis results from episodes of conjunctivitis causing contractile scarring on the palpebral conjunctival surface the upper lid, turning the eyelid and lashes inward so that they can scratch the corneal surface and thus can lead to corneal opacification and vision loss. [6]. Therefore, TT surgery is a key strategy for avoiding blindness and visual impairment from trachoma [5]. Posterior Tarsal Rotation (PLTR) and Bilamellar Tarsal Rotation (BLTR) are WHO-endorsed alternative surgical techniques for TT surgery [7] A prior randomized head-to-head clinical trial found a 43% reduction in the incidence of postoperative TT (PTT) with PLTR compared with BLTR [8], a difference which was sustained through four years after surgery [9]. Even though that trial's surgeons underwent extensive training, practice and re-standardization in both procedures, they had more prior experience of PLTR than BLTR, raising the possibility of performance bias favoring PLTR. Following this result, the WHO has recommended that new TT surgeons be trained in the PLTR technique, but that established surgeons should continue using the technique they prefer [10]. Another recently published clinical trial assessing a third technique vs PLTR and BLTR [11] found 1.2-fold worse (p = 0.058) adjusted odds of postoperative TT with BLTR. In subset analyses, surgeons newly trained in PLTR and BLTR did significantly better with PLTR (1.48-fold higher odds of PTT with BLTR, 95% CI: 1.11-1.97), but surgeons previously trained in BLTR did not (0.83-fold higher odds of PTT with BLTR, 95% CI: 0.58-1.21). The third technique was not found to be an improvement [11].

Because there is residual uncertainty in the trachoma community as to whether we should continue using both the PLTR and BLTR techniques, we undertook an observational comparison of the alternative techniques during an ongoing randomized clinical trial evaluating a separate issue—comparison of ancillary treatment with fluorometholone 0.1% twice daily vs. placebo to evaluate the hypothesis that ancillary fluorometholone 0.1% reduces the incidence of PTT. In our study, surgeons were instructed to continue using one of the two techniques they preferred and regularly practiced as per the WHO recommendation. Because the parent trial's treatment assignment was stratified by surgeon, the circumstances of the parent trial gave us the opportunity evaluate one-year PTT outcome by the use of PLTR or BLTR surgical technique, reported here.

## Methods

### Ethics statement

This study was approved at outset, and approval was regularly extended, as required by each of the following Institutional Review Boards (IRBs): Ethiopian Food and Drug Administration [reference number 02\25\30\103]; Ethiopian National Research Ethics Review Committee [reference number 17\152\45\23]; Mass General Brigham IRB [protocol #:

2019P002286, the IRB for the principal investigator/grant recipient/sponsor]; and London School of Hygiene and Tropical Medicine IRB [reference number 17937 - 02, the IRB for the study Vice-Chair].

All participants provided written informed consent. Consent materials were available in the three languages appropriate to the region, most using the Oromo language.

## Study design

This observational cohort analysis evaluated the incidence of PTT for eyes of 1141 consecutive participants enrolled in the parent clinical trial, in relation to use of the PLTR or BLTR TT surgical technique. The participants studied here represent the first 1141 enrolled in the FLuorometholone as Adjunctive MEdical Therapy for TT Surgery (FLAME) Trial (ClinicalTrials.gov registration number: NCT04149210). The methods of the parent FLAME Trial are described in the study's design paper [10], and additional details of the protocol are posted at https://www.clinicaltrials.gov/study/NCT04149210 [accessed on January 17, 2025]. In brief, the FLAME Trial is a large-scale prospective field trial of participants with TT who had decided to undergo TT surgery on one or both upper eyelids at participating outreach sites in the rural parts of the Jimma zone, Oromia region, Ethiopia. Eligible persons are enrolled and randomized to perioperative treatment with topical fluorometholone 0.1% eyedrops or placebo twice daily for four weeks in the operated eye(s). Participants are followed through one year for the primary outcome of PTT. Surgery is done by trained nurses who, prior to the FLAME Trial, were certified by the Jimma Zone/Oromia Regional Health Bureau/Fred Hollows Foundation ongoing TT surgery outreach program. Randomized study treatment is stratified by surgeon, using a permuted block size of 2 or 4, to ensure that treatment allocation is close to even within each surgeon's cases. This approach also accomplishes stratification by Kebele (geographic subdivision), because surgeons only operate in areas close to where they reside and routinely conduct TT surgery. Because the FLAME Trial is ongoing, and treatment assignment is balanced by surgeon-stratified randomization, we do not adjust for treatment assignment in this analysis.

## Participants

Eligibility criteria (inclusion and exclusion criteria) for FLAME Trial enrollment are listed in S1 Table. In brief, eligible participants are 15 years or older (old enough to have surgery with local anesthesia), have primary TT that is going to be operated in the program, and have neither contraindications to trial treatments nor indications to use corticosteroids for some other reason.

## Procedures

Because the intent of the FLAME Trial is to generalize results to ongoing programs, surgeons use either PLTR or BLTR for participant surgeries based on their clinical preference, per WHO recommendation [10]; they do not receive any training in the technique other than as per routine programmatic practice. PTT is determined using clinical exam by trained study team nurses, defined as: one or more lashes touching the globe in an eye operated for TT; and/or; history of repeat TT surgery; and/or evidence of epilation on clinical examination. Surgery and perioperative treatment are done according to standard programmatic operating procedures of the ongoing surgical program.

## Measurement of primary and secondary outcomes, and covariates

Follow-up of participants was conducted at four weeks, six months and 12 months after enrollment/surgery for ascertainment of the primary and additional outcomes. Additional variables relevant to this analysis at baseline and within one year include: age; body mass index; gender; literacy; education; extent of ocular pain/discomfort; presenting logMAR visual acuity; cataract status; number and location of lashes touching the globe; evidence of epilation; presence and extent of entropion; extent of upper eyelid trachomatous scarring; and type of TT surgery performed (PLTR or BLTR). Adverse

events are monitored prospectively. Because the primary adverse outcomes of interest for this analysis also were collected prospectively, we have relied on the prospective data rather than potentially inconsistent adverse event reporting for evaluating eyelid closure defect, eyelid contour abnormality, eyelid granuloma, eyelid notch, and/or overcorrection. Study nurses collecting the clinical data were standardized to a supervising nurse by grading the same patients one after another repeatedly throughout the trial to assess for potential drift in gradings (which was not observed).

## Statistical analysis

We compared baseline participant characteristics between PLTR and BLTR groups using two-sample $t$-tests for continuous measures and $\chi^2$ tests for categorical measures. We compared baseline ocular characteristics between the PLTR and BLTR groups using generalized estimating equations (GEE) to account for the inter-eye correlation for participants with bilateral surgery. The incidences of PTT by each of the follow-up visits (week 4, months 6 and 12) were calculated and compared between the PLTR and BLTR groups by calculating an odds ratio (OR) and its 95% confidence intervals (95% CI) using GEE. To account for possible confounding from imbalance in baseline characteristics, multivariable logistic regression analyses were performed for comparing the cumulative incidence of PTT during one year's follow-up between the PLTR and BLTR groups. In addition, to accommodate longitudinal changes in PTT, longitudinal analysis of PTT using repeated measures multivariable logistic regression model was performed. Adjusted odds ratios (aOR) with 95% CI for comparing the PLTR and BLTR groups and for the other baseline risk factors were calculated from these multivariable analyses that accounted for the inter-eye correlation from GEE. The incidences of ocular surgical adverse events between the PLTR and BLTR groups were compared using the logistic regression model, accounting for inter-eye correlation using GEE. All statistical analyses were performed in SAS v9.4 (SAS Institute Inc., Cary, NC), and two-sided $p < 0.05$ was considered statistically significant.

## Results

This analysis included 1,141 participants (1,533 eyes) who had completed the 12 months' FLAME Trial follow-up by October 30, 2023. Among the 1,533 eyes, 885 and 678 respectively had TT surgery using the PLTR or BLTR techniques respectively (Tables 1 and 2). At baseline, TT-related characteristics tended to be worse in the PLTR compared to the BLTR group, with higher levels of pain or discomfort, more epilation, more entropion, greater conjunctival scarring severity, and worse visual acuity in affected eyes (all statistically significant, see Table 2). Compared with BLTR surgeons, PLTR surgeons were more numerous and had less experience in terms of number of surgeries done, number of surgeries done in the prior year, and time since training (Table 3).

Seventy (8.2%) vs 146 (21.4%; risk difference = -0.13, 95% CI: -0.169, -0.097) study eyes in the PLTR and BLTR groups respectively developed PTT prior to or at the one-year visit ($p < 0.001$; Fig 1). The cross-sectional incidence of PTT was lower in the PLTR group than the BLTR group at four weeks (1.3% vs. 3.1%), six months (6.3% vs. 13.8%), and 12 months (5.3% vs. 17.7%; S1 Fig, with corresponding significantly lower odds ratios (OR) 0.41 (95% CI: 0.18-0.92), 0.42 (95% CI: 0.29-0.62) and 0.26 (95% CI: 0.17, 0.38) respectively (Table 4). The mean number of lashes touching the eye and proportion with entropion also were correspondingly lower in the PLTR group than the BLTR group at all follow-up visits (Table 4).

Multivariable logistic regression modeling of the incidence of PTT (Table 5) that accounted for the baseline characteristics and inter-eye correlation confirmed a lower cumulative 12-month incidence of PTT in the PLTR group than in the BLTR group (adjusted odds ratio (aOR)=0.27; 95% CI, 0.19, 0.39). The model adjusted for baseline risk factors expected to be predictive of PTT, including higher age (aOR per each 10 years older = 1.17; 95% CI, 1.04, 1.31); number of trichiatic lashes or epilation (6 or more vs epilation, aOR=0.87, 95% CI: 0.51, 1.48; 1–5 lashes vs epilation, aOR=0.61, 95% CI: 0.36, 1.03; overall $p = 0.09$); increased severity of entropion (showing an approximate dose-response relationship, overall $p = 0.0502$); and extent of upper eyelid trachomatous scarring (no significantly associated, overall $p = 0.73$).

**Table 1. Baseline characteristics of participants undergoing trachomatous trichiasis surgery in the FLuorometholone as Adjunctive MEdical Therapy for TT Surgery (FLAME) Trial\*.**

| | PLTR | BLTR | p value |
|---|---|---|---|
| **Participants** | N = 634 | N = 507 | |
| Age in years, mean (SD) | 48.0 (15.2) | 47.2 (15.3) | 0.38 † |
| BMI, mean (SD) | 18.8 (2.3) | 19.5 (2.6) | <0.001 † |
| Gender, n (%) female | 441 (69.6%) | 359 (70.8%) | 0.65 ‡ |
| Reading, n (%) unable | 571 (90.1%) | 450 (88.8%) | 0.47 ‡ |
| No formal education, n (%) | 572 (90.2%) | 451 (89.0%) | 0.49 ‡ |
| Pain or Discomfort, n (%) | | | |
| None | 154 (24.3%) | 92 (18.1%) | <0.001 ‡ |
| Slight | 180 (28.4%) | 156 (30.8%) | |
| Moderate | 113 (17.8%) | 160 (31.6%) | |
| Severe | 142 (22.4%) | 72 (14.2%) | |
| Extreme | 45 (7.1%) | 27 (5.3%) | |
| OSDI Ocular symptoms, mean (SD) | 52.1 (31.6) | 55.1 (30.6) | 0.11 † |
| OSDI Total score, mean (SD) | 41.3 (25.6) | 42.8 (23.4) | 0.29 † |

\* PLTR = posterior lamellar tarsal rotation; BLTR = bilamellar tarsal rotation; SD = standard deviation; BMI = body mass index; OSDI = Ocular Surface Disease Index (reference 10); Treatment assignment is not studied as a covariate because it is stratified by surgeon (see text).

† Independent two sample t-test.

‡ Chi-squared test.

Overall adverse surgical outcomes (Table 6) were less common in the PLTR than the BLTR group respectively at the four week time point (27.2% vs 37.0%, p<0.001)—including eyelid closure defects (0.7% vs. 1.8%, p=0.08), eyelid contour abnormality (24.5% vs. 32.8%, p=0.001), eyelid notch (15.9% vs 23.5%, p=0.001), overcorrection (8.0% vs 16.4%, p<0.001) and eyelid granuloma (2.9% vs 4.0%, p=0.32). However, many of these early adverse outcomes tended to be mild, and most resolved over time. Infections diagnosed at this time point (2.93% and 2.81% respectively, p=0.90) occurred with similar frequency in the two groups. By six months (overall=9.2% vs 9.2%, p=1.00) and 12 months (3.6% vs 3.4%, p=0.81), the overall adverse outcome prevalence was lower and similarly distributed in the PLTR and BLTR groups. At 12 months, in the PLTR and BLTR groups respectively, the cross-sectional prevalences of eyelid closure defects (0.0% vs. 0.1%, p=not calculable), eyelid contour abnormality (3.3% vs. 3.1%, p=0.85), eyelid notch (2.3% vs 2.2%, p=0.88), overcorrection (0.6% vs 0.6%, p=0.99) and eyelid granuloma (0.1% vs 0.4%, p=0.25) were not significantly different. Infections at the six (0.24% vs 0.45%, p=0.50) and 12 month (0.58% vs 0.15%, p=0.25) time points were rare and occurred with similar frequency in the two groups respectively.

## Discussion

In our "real world" study of eyes undergoing PLTR or BLTR in an ongoing TT surgery outreach program in rural Ethiopia, we observed that cumulative PTT was less frequent with PLTR (8.2%) than BLTR (21.4%) by 12 months. Adjustment for baseline TT severity factors, which were worse in the PLTR group, further increased the advantage of PLTR over BLTR surgery with respect to PTT incidence. A previous head-to-head clinical trial by Habtamu, et al, found that PLTR was superior to BLTR for avoiding PTT [8,9], but had been criticized in that participating surgeons may have been more familiar with the PLTR than the BLTR procedure (although they had completed at least 100 or more of each procedure prior to the trial). Compared to Habtamu et al's trial [7], in our study, where surgeons used their preferred procedure of the two, results were similar or perhaps favored PLTR even more. Thus, our results confirmed the results of that trial [7], suggesting that PLTR is more effective for TT surgery in terms of avoiding post-operative

**Table 2. Baseline characteristics of eyes undergoing trachomatous trichiasis surgery in the FLuorometholone as Adjunctive MEdical Therapy for TT Surgery (FLAME) Trial\*.**

| | PLTR | BLTR | p value |
|---|---|---|---|
| **Eyes** | N = 855 | N = 678 | |
| logMAR VA, mean (SD) | 0.53 (0.52) | 0.45 (0.39) | 0.002 † |
| Cataract, n (%) | 247 (28.9%) | 147 (21.7%) | 0.007 † |
| Number of upper lid lashes touching globe, n (%) | | | |
| Zero (epilating) | 134 (15.7%) | 65 (9.6%) | 0.02 † |
| 1–5 | 518 (60.6%) | 438 (64.6%) | |
| 6 or more | 203 (23.7%) | 175 (25.8%) | |
| Evidence of epilation, n (%) | 145 (17.0%) | 74 (10.9%) | 0.007 † |
| Presence of entropion, n (%) | 780 (91.2%) | 555 (81.9%) | <0.001 † |
| Severity of entropion, n (%) | | | |
| None | 75 (8.8%) | 123 (18.1%) | <0.001 † |
| Mild | 307 (35.9%) | 252 (37.2%) | |
| Moderate | 183 (21.4%) | 149 (22.0%) | |
| Severe | 212 (24.8%) | 126 (18.6%) | |
| Complete | 78 (9.1%) | 28 (4.1%) | |
| Conjunctivalization of upper lid margin, n (%) | | | |
| None | 11 (1.3%) | 22 (3.2%) | <0.001 † |
| Mild | 61 (7.1%) | 111 (16.4%) | |
| Moderate | 281 (32.9%) | 343 (50.6%) | |
| Severe | 502 (58.7%) | 202 (29.8%) | |
| Upper eyelid trachomatous scarring, n (%) | | | |
| None | 51 (6.0%) | 84 (12.4%) | <0.001 † |
| Mild | 37 (4.3%) | 83 (12.2%) | |
| Moderate | 301 (35.2%) | 322 (47.5%) | |
| Severe | 466 (54.5%) | 189 (27.9%) | |

\* PLTR = posterior lamellar tarsal rotation; BLTR = bilamellar tarsal rotation; logMAR = log10(mean angle of resolution); Treatment assignment is not studied as a covariate because it is stratified by surgeon (see text); Upper eyelid trachomatous scarring is defined as: None: No scarring on the conjunctiva; Mild: fine scattered scars on the upper tarsal conjunctiva, or scars on the other parts of the conjunctiva; Moderate: more severe scarring but without shortening or distortion of the upper tarsus; Severe: scarring with distortion of the upper tarsus.

† From generalized estimating equations.

TT. The difference in both studies was large and clinically important, and is biologically plausible given the wedge of tissue left in place with the PLTR technique that likely resists progressive postoperative scarring causing inward rotation postoperatively [8]. Despite BLTR surgeons having had more experience and PLTR surgeons often being trained in the last six months (as per the WHO recommendation that new surgeons be trained in PLTR), PLTR outcomes were superior. The new trial by Gower, et al. [11], tended to show the same pattern, but its overall results did not as strongly favor PLTR as in Habtamu et al's and our study and the result was not statistically significant in favor of PLTR. However, the subset analysis for their surgeons newly trained in both PLTR and BLTR clearly favored PLTR as having more favorable results for PTT incidence. However, surgeons previously trained in BLTR did not perform better with PLTR after being trained in it.

PLTR also had an early advantage over BLTR in prospectively assessed common postoperative adverse outcomes, significantly favoring PLTR at four weeks postoperatively. However, by six and 12 months postoperatively, the cross-sectional prevalence of these postoperative adverse outcomes was small and similar across surgical techniques, despite

**Table 3. Comparison between PLTR vs. BLTR surgeons in their experience of TT surgeries pre-FLAME\*.**

| | PLTR | BLTR |
|---|---|---|
| **Surgeons** | **N = 23 surgeons** | **N = 7 surgeons** |
| Count of Surgeries Done Pre-FLAME | | |
| Mean (Standard Deviation [SD]) | 139.7 (144.5) | 862.4 (538.3) |
| Median (Quartile 1, Quartile 3) | 62 (40, 309) | 745 (500, 1000) |
| Number of Surgeries *per Year* Pre-FLAME | | |
| Mean (SD) | 95.7 (94.3) | 198.9 (97.0) |
| Median (Quartile 1, Quartile 3) | 47 (23, 155) | 210 (117, 297) |
| Number of Years of Experience in Performing TT Surgery pre-FLAME | | |
| Mean (SD) | 0.5 (0.5) | 6.7 (0.7) |
| Median (Quartile 1, Quartile 3) | 0.2 (0.1, 0.7) | 7.0 (6.0, 7.3) |

\* PLTR = posterior lamellar tarsal rotation; BLTR = bilamellar tarsal rotation; TT = trachomatous trichiasis; FLAME = FLuorometholone as Adjunctive MEdical Therapy for TT Surgery (FLAME) Trial.

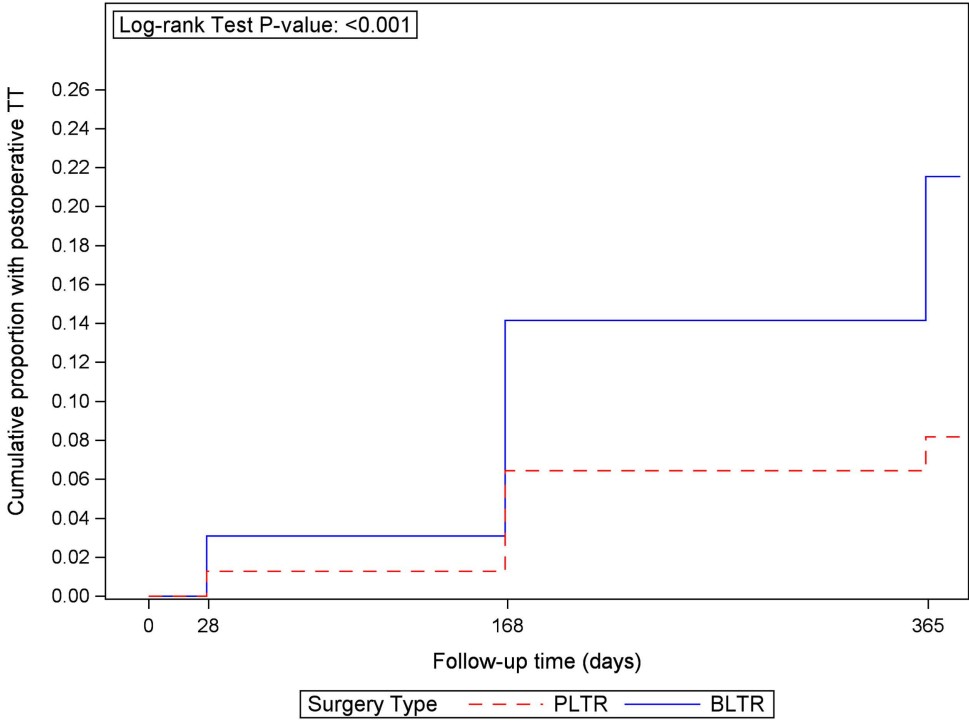

**Fig 1. Cumulative incidence of postoperative trachomatous trichiasis (TT) during follow-up after bilamellar tarsal rotation (BLTR) or posterior lamellar tarsal rotation (PLTR) surgery.**

a lack of interventions in the large majority of cases. The early difference likely reflects differences in the early postoperative clinical course of the alternative surgical procedures and does not have an important impact on choosing which is the more appropriate procedure for programmatic use given that both procedures had similarly few adverse postoperative outcomes in the long run.

**Table 4. Primary and secondary outcomes of BLTR vs PLTR surgery, FLuorometholone as Adjunctive MEdical therapy for TT surgery (FLAME) trial*.**

| | PLTR | | BLTR | | Odds ratios (95% CI) | p value† | Risk difference (95% CI) |
|---|---|---|---|---|---|---|---|
| | N = 855 | | N = 678 | | | | |
| Postoperative TT, n (%) | | | | | | | |
| 4 weeks | 11 | 1.3% | 21 | 3.1% | 0.41 (0.18,0.92) | 0.03 | -0.018 (-0.033, -0.003) |
| 6 months | 53 | 6.3% | 93 | 13.8% | 0.42 (0.29,0.62) | <0.001 | -0.075 (-0.106, -0.044) |
| 12 months | 45 | 5.3% | 120 | 17.7% | 0.26 (0.17,0.38) | <0.001 | -0.124 (-0.157, -0.092) |
| Any time during FU | 70 | 8.2% | 146 | 21.5% | 0.32 (0.23,0.45) | <0.001 | -0.133 (-0.169, -0.097) |
| Number of lashes touching eye, mean (SD)# | | | | | | | |
| 4 weeks | 0.0 (0.3) | | 0.1 (0.4) | | | 0.13 | |
| 6 months | 0.1 (0.6) | | 0.3 (1.2) | | | <0.001 | |
| 12 months | 0.1 (0.6) | | 0.4 (1.2) | | | <0.001 | |
| Presence of entropion, n (%) | | | | | | | |
| 4 weeks | 9 | 1.1% | 18 | 2.7% | 0.39 (0.16,0.93) | 0.03 | |
| 6 months | 69 | 8.3% | 111 | 16.5% | 0.46 (0.32,0.65) | <0.001 | |
| 12 months | 70 | 8.2% | 141 | 20.8% | 0.34 (0.24,0.48) | <0.001 | |

* PLTR = posterior lamellar tarsal rotation; BLTR = bilamellar tarsal rotation; TT = trachomatous trichiasis; CI = confidence interval; E = number of eyes; FU = follow-up; SD = standard deviation.

† From generalized estimating equations.

# among those with PTT, the count of lashes was similar between the PLTR and BLTR groups respectively, at 4 weeks (2.5 vs 2.1, p = 0.5), 6 months (1.8 vs 2.4, p = 0.06) and 12 months (2.0 vs 2.4, p = 0.3).

As expected, initially more severe upper eyelid entropion and trichiasis as well as greater age tended to be associated with a higher incidence of PTT. Because these factors were worse at baseline in the PLTR group, adjustment for these factors further amplified the difference in PTT to 73% lower odds with PLTR than BLTR.

The study had strengths including a large sample size allowing for precise estimates of risk, and that PLTR surgeons were *not* more experienced than BLTR surgeons (addressing a potential criticism regarding the trial of Habtamu et al. [7]), with each surgeon using their preferred procedure. Data were collected prospectively as part of a parent randomized controlled trial studying a separate issue.

Potential limitations of the study include that it was non-randomized for comparison of PLTR and BLTR. Potential limitations of a non-randomized study of treatment include potential indications-for-treatment biases, and a lack of balance of unknown confounders. Indication-for-treatment bias is unlikely to have been a major problem in this analysis because the decision of which surgeon would operate cases was made before cases were identified. Furthermore, the major result confirms similar results from Habtamu et al's clinical trial [8] and results in the same direction from Gower, et al's trial, especially among newly trained surgeons [11]. Imbalance of unobserved confounders cannot be ruled out; but in a large study with large differences between groups, it is unlikely that unrecognized confounding would qualitatively change the results. While the parent trial's study treatments (fluorometholone 0.1% vs placebo) are hypothesized to affect the incidence of post-operative TT, any effects likely would be balanced across surgeons because randomization was stratified by surgeon and the number of participants was large compared to the number of surgeons. While only participants who completed follow-up in the period studied were included, it is unlikely this group differed from the group enrolled because follow-up at 12 months was more than 98%. Surgeon training quality may have differed over time but is unlikely to have differed sufficiently to generate the large difference in results observed, especially taking into account similar results reported elsewhere and the biological plausibility of the results observed.

**Table 5. Crude and adjusted relative odds for ever having postoperative trachomatous trichiasis (PTT) in 12 months by type of surgery (BLTR vs PLTR) and other factors in the FLuorometholone as Adjunctive MEdical therapy for TT surgery (FLAME) trial\*.**

| Characteristics of Eyes | Postoperative TT | | | Crude | | Adjusted | |
|---|---|---|---|---|---|---|---|
| | Eyes Studied, E= | No (%) | Yes (%) | Odds Ratio (95% CI) | p | Odds Ratio (95% CI) | p |
| Age per 10 years | 1533 | Mean = 46.8 (SD = 14.9) | Mean = 50.0 (SD = 15.5) | 1.15** (1.03,1.28) | 0.01 | 1.17** (1.04,1.31) | 0.006 |
| Number of upper lid lashes touching globe/Epilating | | | | | | | |
| Zero (epilating) | 199 | 165 (82.9%) | 34 (17.1%) | Ref | 0.005 | Ref | 0.09 |
| 1-5 | 956 | 844 (88.3%) | 112 (11.7%) | 0.64 (0.40,1.05) | | 0.61 (0.36,1.03) | |
| 6 or more | 378 | 308 (81.5%) | 70 (18.5%) | 1.10 (0.65,1.86) | | 0.87 (0.51,1.48) | |
| Severity of entropion | | | | | | | |
| None | 198 | 173 (87.4%) | 25 (12.6%) | Ref | 0.03 | Ref | 0.0502 |
| Mild | 559 | 491 (87.8%) | 68 (12.2%) | 0.96 (0.58,1.59) | | 1.31 (0.77,2.22) | |
| Moderate | 332 | 293 (88.3%) | 39 (11.7%) | 0.92 (0.53,1.61) | | 1.20 (0.68,2.12) | |
| Severe | 338 | 276 (81.7%) | 62 (18.3%) | 1.55 (0.93,2.61) | | 2.06 (1.18,3.60) | |
| Total | 106 | 84 (79.2%) | 22 (20.8%) | 1.81 (0.92,3.56) | | 2.41 (1.17,4.96) | |
| Upper eyelid trachomatous scarring, % | | | | | | | |
| None | 135 | 114 (84.4%) | 21 (15.6%) | Ref | 0.65 | Ref | 0.73 |
| Mild | 120 | 99 (82.5%) | 21 (17.5%) | 1.15 (0.57,2.31) | | 1.03 (0.50,2.15) | |
| Moderate | 623 | 541 (86.8%) | 82 (13.2%) | 0.82 (0.47,1.43) | | 0.82 (0.46,1.48) | |
| Severe | 655 | 563 (86.0%) | 92 (14.0%) | 0.89 (0.51,1.55) | | 0.99 (0.53,1.87) | |
| Type of TT Surgery | | | | | | | |
| BLTR | 678 | 532 (78.5%) | 146 (21.5%) | Ref | <.0001 | Ref | <.0001 |
| PLTR | 855 | 785 (91.8%) | 70 (8.2%) | 0.32 (0.23,0.45) | | 0.27 (0.19,0.39) | |

\* PLTR = posterior lamellar tarsal rotation; BLTR = bilamellar tarsal rotation; TT = trachomatous trichiasis; E = number of eyes; CI = confidence interval; SD = standard deviation; Ref = reference group; PTT refers to incidence of postoperative TT during the 12 month follow-up period. Upper eyelid trachomatous scarring is defined as: None: No scarring on the conjunctiva; Mild: fine scattered scars on the upper tarsal conjunctiva, or scars on the other parts of the conjunctiva; Moderate: more severe scarring but without shortening or distortion of the upper tarsus; Severe: scarring with distortion of the upper tarsus.

\*\*Odds ratios for age are counted for every ten years of increased age.

A multivariable repeated measures logistic regression model for longitudinal measures of PTT at each follow-up visits (week 4, months 6 and 12) yielded similar results (S2 Table).

In summary, this analysis of PLTR vs BLTR for trachomatous trichiasis demonstrated a large and clinically important reduction in the incidence of PTT with PLTR corresponding to a 13.2% lower burden of PTT, without an increase in adverse events. The results are confirmed by congruous results of one prior clinical trial [9] and are consistent with results (particularly among newly trained surgeons) in another trial [11]. Adverse outcomes differed little after a transient early advantage for PLTR. Together, these results suggest that PLTR is a preferable procedure to BLTR for trachomatous trichiasis surgery for newly trained surgeons and surgeons already trained in PLTR. Surgeons who already prefer BLTR should continue performing BLTR, based on Gower, et al's results for that group [11].

## Research in context

**Evidence before this study.** The World Health Organization (WHO) endorses two surgical procedures for use in trachomatous trichiasis (TT) programs globally: posterior lamellar tarsal rotation (PLTR also known as Trabut) and bilamellar tarsal rotation BLTR). A head-to-head randomized field clinical trial comparing the two techniques in upper lid TT found 43% lower postoperative TT with the PLTR procedure. Even though that trial's surgeons underwent extensive

**Table 6. Incidence of adverse outcomes by type of trachomatous trichiasis (TT) surgery (posterior lamellar tarsal rotation [PLTR] vs bilamellar tarsal rotation [BLTR]), in the FLuorometholone as Adjunctive MEdical therapy for TT surgery (FLAME) trial*.**

| Adverse outcomes at four weeks, n (%) | PLTR N = 853 | BLTR N = 676 | P value* |
|---|---|---|---|
| *Adverse outcome incidence* | 232 (27.2%) | 250 (37.0%) | <0.001 |
| Eyelid closure defect | 6 (0.7%) | 12 (1.8%) | 0.08 |
| Eyelid contour abnormality | 209 (24.5%) | 222 (32.8%) | 0.001 |
| Eyelid notch | 20 (2.3%) | 15 (2.2%) | 0.88 |
| Eyelid granuloma | 25 (2.9%) | 27 (4.0%) | 0.32 |
| Overcorrection | 68 (8.0%) | 111 (16.4%) | <0.001 |
| **Adverse outcomes at six months, n (%)** | **PLTR N = 835** | **BLTR N = 673** | **P value*** |
| *Adverse outcome incidence* | 77 (9.2%) | 62 (9.2%) | 1 |
| Eyelid closure defect | 0 (0.0%) | 1 (0.1%) | |
| Eyelid contour abnormality | 63 (7.5%) | 38 (5.6%) | 0.17 |
| Eyelid notch | 20 (2.3%) | 15 (2.2%) | 0.88 |
| Eyelid granuloma | 13 (1.6%) | 22 (3.3%) | 0.053 |
| Overcorrection | 17 (2.0%) | 15 (2.2%) | 0.81 |
| **Adverse outcomes at 12 months, n (%)** | **PLTR N = 855** | **BLTR N = 678** | **P value*** |
| *Adverse outcome incidence* | 31 (3.6%) | 23 (3.4%) | 0.81 |
| Eyelid closure defect | 0 (0.0%) | 1 (0.1%) | |
| Eyelid contour abnormality | 28 (3.3%) | 21 (3.1%) | 0.85 |
| Eyelid notch | 20 (2.3%) | 15 (2.2%) | 0.88 |
| Eyelid granuloma | 1 (0.1%) | 3 (0.4%) | 0.25 |
| Overcorrection | 5 (0.6%) | 4 (0.6%) | 0.99 |

**Eyelid closure defect:** Unable to fully close the eyelids leaving part of the globe visible when the participant is asked to gently close the eye lids.

**Eyelid contour abnormality:** The distortion of the normal eye lid curvature(contour) to notching or wavey eye lid margin shape. The severity is determined by the vertical deviation of the eye lid margin from the natural contour or the extent of the horizontal eye lid margin involvement.

**Eyelid notch:** A dimple or groove on the upper eye lid margin the severity depends on the vertical deviation of the eye lid margin from the normal eye lid contour line. It is a subset of eyelid contour abnormality. Eyelid notch is a subset of Eyelid contour abnormalities.

**Eyelid granuloma:**

**Overcorrection:** The over rotation of the upper eyelid margin (more than 2 mm) after trachomatous trichiasis surgery.

training, practice and re-standardization in both procedures, they had more prior experience of PLTR than BLTR, raising the possibility of performance bias favoring PLTR. Following this result, the WHO has recommended that new TT surgeons be trained in the PLTR technique, but that established surgeons should continue using the technique they prefer. A second recent trial assessing a third technique vs PLTR and BLTR found 1.2-fold worse (p = 0.058) adjusted odds of postoperative TT with BLTR, but in subset analyses surgeons newly trained in PLTR and BLTR did significantly better with PLTR but surgeons previously trained in BLTR did not. The third technique was not found to be an improvement.

**Added value of this study.** Because there is residual uncertainty in the trachoma community about whether PLTR is a superior procedure, we compared PTLR vs BLTR in a large ongoing randomized field trial evaluating another issue, nested within a large ongoing TT program in which surgeons used PTLR or BLTR—whichever procedure they preferred. The parent trial aims to compare alternative treatments in a "real world" context, so no additional training was given other than as routine within the ongoing TT program. Despite greater severity of baseline characteristics in the PLTR group and less experience among the PLTR surgeons than the BLTR surgeons, PLTR was associated with much

less cumulative postoperative TT by one year: 8.2% (PLTR) vs 21.4% (BLTR). The adjusted odds ratio favored PTLR (0.27, 95% confidence interval: 0.19-0.39). Adverse outcomes were similar in the two groups at six and 12 months after surgery.

**Implications of all the available evidence.** These real-world data confirm a prior trial's primary result, suggesting that using PLTR instead of BLTR reduces postoperative TT incidence to a clinically important degree without increasing adverse outcomes. Together with the prior randomized trial [7,8], these results suggest that PLTR is a preferable procedure to BLTR for trachomatous trichiasis surgery, with the exception of surgeons already trained in BLTR who should continue to be allowed to do BLTR if they prefer it.

## Supporting information

**S1 Fig. Cross-sectional prevalence of postoperative trachomatous trichiasis (TT) at follow-up visits by type of surgery (posterior lamellar tarsal rotation [PLTR] or bilamellar tarsal rotation [BLTR]).**
(TIFF)

**S1 Table. Eligibility Criteria for the FLuorometholone as Adjunctive MEdical Therapy for Trachomatous Trichiasis Surgery (FLAME) Trial, from whence the first 1141 consecutive participants enrolled were studied in this analysis.**
(DOCX)

**S2 Table. Longitudinal Analysis for Crude and Adjusted Relative Odds for Postoperative Trachomatous Trichiasis (PTT) by Type of Surgery (BLTR vs PLTR), Visit and Baseline Factors in the FLuorometholone as Adjunctive MEdical Therapy for TT Surgery (FLAME) Trial.**
(DOCX)

**S3 Table. Source data supporting the submission can be found in this supplemental file.**
(XLSX)

**S1 File. Acknowledgments.** FLAME Research Group Members, Data and Safety Monitoring Committee Members, and National Eye Institute Program Officers, as of January 29, 2025.
(DOCX)

**S1 Data. Description of location of manuscript source data for tables and figures.**
(DOCX)

## Acknowledgments

The authors acknowledge the valuable contributions of National Eye Institute Program Officers Sangeeta Bhargava, PhD, and Jimmy Le, ScD to the FLAME Trial.

## Author contributions

**Conceptualization:** John H. Kempen, Aida Abashawl, Wondu Alemayehu, Maureen G. Maguire, Matthew J. Burton, Gui-shuang Ying.

**Data curation:** John H. Kempen, Yineng Chen, Aida Abashawl, Ahlam Awad Mohammed, Aemero Abateneh Mengesha, Fangming Jin, Maureen G. Maguire, Gui-shuang Ying.

**Formal analysis:** Yineng Chen, Fangming Jin, Maureen G. Maguire, Gui-shuang Ying.

**Funding acquisition:** John H. Kempen, Aida Abashawl, Sarity Dodson, Vatinee Y. Bunya, Matthew J. Burton, Gui-shuang Ying.

**Investigation:** John H. Kempen, Yineng Chen, Aida Abashawl, Ahlam Awad Mohammed, Sarity Dodson, Wondu Alemayehu, Alemu Gemechu, Aemero Abateneh Mengesha, Dereje Adugna Kumsa, Tony Succar, Kathleen McWilliams, Maureen G. Maguire, Matthew J. Burton, Gui-shuang Ying.

**Methodology:** John H. Kempen, Aida Abashawl, Ahlam Awad Mohammed, Sarity Dodson, Wondu Alemayehu, Aemero Abateneh Mengesha, Fangming Jin, Vatinee Y. Bunya, Matthew J. Burton, Gui-shuang Ying.

**Project administration:** John H. Kempen, Aida Abashawl, Sarity Dodson, Vatinee Y. Bunya, Gui-shuang Ying.

**Resources:** John H. Kempen, Yineng Chen, Aida Abashawl, Ahlam Awad Mohammed, Sarity Dodson, Wondu Alemayehu, Alemu Gemechu, Aemero Abateneh Mengesha, Dereje Adugna Kumsa, Tony Succar, Kathleen McWilliams, Vatinee Y. Bunya, Maureen G. Maguire, Matthew J. Burton, Gui-shuang Ying.

**Software:** John H. Kempen, Aida Abashawl, Ahlam Awad Mohammed, Sarity Dodson, Gui-shuang Ying.

**Supervision:** John H. Kempen, Aida Abashawl, Ahlam Awad Mohammed, Sarity Dodson, Wondu Alemayehu, Alemu Gemechu, Aemero Abateneh Mengesha, Dereje Adugna Kumsa, Tony Succar, Kathleen McWilliams, Maureen G. Maguire, Matthew J. Burton, Gui-shuang Ying.

**Validation:** John H. Kempen, Yineng Chen, Aida Abashawl, Ahlam Awad Mohammed, Alemu Gemechu, Aemero Abateneh Mengesha, Kathleen McWilliams, Maureen G. Maguire, Gui-shuang Ying.

**Visualization:** John H. Kempen, Yineng Chen, Fangming Jin, Maureen G. Maguire, Gui-shuang Ying.

**Writing – original draft:** John H. Kempen.

**Writing – review & editing:** John H. Kempen, Yineng Chen, Aida Abashawl, Ahlam Awad Mohammed, Sarity Dodson, Wondu Alemayehu, Alemu Gemechu, Aemero Abateneh Mengesha, Dereje Adugna Kumsa, Tony Succar, Kathleen McWilliams, Fangming Jin, Vatinee Y. Bunya, Maureen G. Maguire, Matthew J. Burton, Gui-shuang Ying.

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
