## [Decision Letter · Decision Letter 0]

8 Nov 2024

PNTD-D-24-01260Outcomes of Posterior Lamellar Tarsal Rotation vs Bilamellar Tarsal Rotation for Trachomatous TrichiasisPLOS Neglected Tropical Diseases Dear Dr. Kempen, Thank you for submitting your manuscript to PLOS Neglected Tropical Diseases. After careful consideration, we feel that it has merit but does not fully meet PLOS Neglected Tropical Diseases's publication criteria as it currently stands. Therefore, we invite you to submit a revised version of the manuscript that addresses the points raised during the review process. Please submit your revised manuscript within 60 days Jan 07 2025 11:59PM. If you will need more time than this to complete your revisions, please reply to this message or contact the journal office at plosntds@plos.org. Please include the following items when submitting your revised manuscript:* A rebuttal letter that responds to each point raised by the editor and reviewer(s). You should upload this letter as a separate file labeled 'Response to Reviewers '. This file does not need to include responses to any formatting updates and technical items listed in the 'Journal Requirements' section below.* A marked-up copy of your manuscript that highlights changes made to the original version. You should upload this as a separate file labeled 'Revised Manuscript with Track Changes '.* An unmarked version of your revised paper without tracked changes. You should upload this as a separate file labeled 'Manuscript '. If you would like to make changes to your financial disclosure, competing interests statement, or data availability statement, please make these updates within the submission form at the time of resubmission. Guidelines for resubmitting your figure files are available below the reviewer comments at the end of this letter. We look forward to receiving your revised manuscript. Kind regards, Aleksandra Inic-KanadaGuest EditorPLOS Neglected Tropical Diseases Stuart BlacksellSection EditorPLOS Neglected Tropical Diseases

Shaden Kamhawi

co-Editor-in-Chief

Paul Brindley

co-Editor-in-Chief

 **Journal Requirements:** **Additional Editor Comments (if provided):** Dear Authors,

While this is an important paper, some significant limitations need to be addressed. In addition to the limitations listed in the public section: There is a new clinical trial that reports on the nuances of this issue - timing of surgeon training, which procedure was taught first, etc all matter.

https://journals.plos.org/plosntds/article?id=10.1371/journal.pntd.0012034

Trial registration

Registration number: NCT03100747; ClinicalTrials.gov

Full study protocol available at (https://doi.org/10.15139/S3/QHZXWD)

The article was just published when your manuscript was submitted, so you likely missed it. Now that it’s available, it would be beneficial to address its findings in the current article. Best wishes,Aleksandra Inic-Kanada  **Reviewers' Comments:** Reviewer's Responses to Questions

**Key Review Criteria Required for Acceptance?**

**Methods**

-Are the objectives of the study clearly articulated with a clear testable hypothesis stated?

-Is the study design appropriate to address the stated objectives?

-Is the population clearly described and appropriate for the hypothesis being tested?

-Is the sample size sufficient to ensure adequate power to address the hypothesis being tested?

-Were correct statistical analysis used to support conclusions?

-Are there concerns about ethical or regulatory requirements being met?

Reviewer #1: Objectives clearly stated, study design and sample size clear and appropriate, correct statistical methods used and no ethical or regulatory concerns

Reviewer #2: The authors have articulated a clear, testable hypothesis, and the study design is suitable for addressing the stated objectives. The sample size appears sufficient to provide adequate power for the tested hypothesis, and the statistical analyses used are appropriate, effectively supporting the conclusions.

However, the methodology section would benefit from clearer structure to guide readers on how the study design addresses the research questions. Additionally, providing more detailed information on the population under study would enhance understanding of its appropriateness for the hypothesis.

The current methodology section does not include any information regarding ethical approval or consent. To ensure compliance with ethical standards the authors need to add a statement that addresses the ethical aspects of this research. This should include details on the ethical approval obtained from an appropriate institutional review board or ethics committee, as well as information on informed consent from participants.

Please refer to the attached file for specific comments.

Reviewer #3: The overall study approach is clear. The authors need to provide methods describing how they defined each of the adverse events, specifically eyelid contour abnormality (definitions and position of the eyelid for making the assessment), eyelid not (how does this differ from contour abnormality?), over-correction as well as the outcome "trachomatous scarring".

What sort of training did the clinical outcomes assessors have? How were they standardized? Was kappa testing conducting to confirm that all examiners grade things similarly?

What sort of quality control procedures are in place for evaluating outcomes? Did the authors check for drift over time, etc?

The analysis description suggests that all adverse events were clustered together and analyze as y/n "any" adverse event. Given that different procedures have different rates of each adverse event type, consider individual analyses comparing a given event (e.g. eyelid contour abnormality) between the 2 groups. This is the standard in TT surgery trials.

**Results**

-Does the analysis presented match the analysis plan?

-Are the results clearly and completely presented?

-Are the figures (Tables, Images) of sufficient quality for clarity?

Reviewer #1: Analysis matches the plan, results clearly and completely presented. Figures and tables clear apart from table 1, in which percentages exceeding 100 are shown. This needs to be corrected

Reviewer #2: Please refer to the attached file for specific comments.

Reviewer #3: Table 3: please check total versus last year surgeries for both groups. At present, they are exactly the same, which doesn't make sense given the number of years of surgery reported.

Table 4: By definition, the way the analyses are reported, the BLTR group would have to have a higher number of lashes, since more eyes had TT. This table would be more informative if the data about PTT severity was limited to eyes that had PLTT. Consider just making 1 footnote for gee, in the column heading, instead of having it in every row.

Lines 299-304: This section is unclear. An eyelid can have epilation, trichiatic eyelashes, or both. So it is unclear why the comparisons were made 6 or more vs epilation, 1-5 vs epilation, etc. Typically in TT surgery trials, eyelids are assigned a severity level based on number of eyelashes touching the eye and/or amount of epilation.

line 312 references a lancet website. Is this correct?

Supplemental table 2: Details are needed on the timing of the variables shown - are the rows baseline levels or the amount at follow up?

**Conclusions**

-Are the conclusions supported by the data presented?

-Are the limitations of analysis clearly described?

-Do the authors discuss how these data can be helpful to advance our understanding of the topic under study?

-Is public health relevance addressed?

Reviewer #1: Yes to all. The results of this study are of considerable clinical importance and should lead to a change in the WHO recommendations for trichiasis surgery.

Reviewer #2: Please refer to the attached file for specific comments.

Reviewer #3: Determination cannot be made with regards to whether the conclusions are supported by the data until additional data are provided.

**Editorial and Data Presentation Modifications?**

Reviewer #1: The percentages in table 1 need to be corrected

Reviewer #2: Please refer to the attached file for details.

Reviewer #3: There are a number of editorial suggestions:

There are a number of inconsistencies in terminology, typos, and potential lack of proofreading the submission - for example, line 187 "study's design study paper", line 197 "block, size 2 or 4" (comma in the wrong place), "trachoma-tous" used in all tables, line 238 "one year's follow up", abstract "PTLR" instead of "PLTR", line 166 Posterior Tarsal Rotation (should be Posterior Lamellar Tarsal Rotation), table 4 "percent of have", line 311 "at each follow up visits", etc.

line 150-151: Consider rewording, as "primarily mediated by" suggests that there are multiple components of trachoma that cause blindness, but to this reviewer's knowledge, corneal opacity caused by TT is the necessary pathway for blindness in trachoma.

line 170 what is meant by "a difference which was *qualitatively* sustained"?

line 217 (and elsewhere) "extant" suggests something old but still surviving. Consider a different word that conveys a thriving, active surgical program?

Consider using "participants" or "subjects" instead of "patients"

It is unclear why PLTR and BLTR are referred to as "alternative" procedures. These are the 2 standard surgical procedures for TT.

**Summary and General Comments**

Reviewer #1: This is an important paper which shows conclusively that the PLTR operation for trachomatous trichiasis leads to better long term outcomes than BLTR in real world conditions, even when the surgeons performing PLTR have less experience than those performing BLTR.

Reviewer #2: Kempen et al. present a manuscript that demonstrates the superiority of posterior lamellar tarsal rotation (PLTR) over bilamellar tarsal rotation (BLTR) in the treatment of trachomatous trichiasis, indicating that PLTR is a more effective procedure for the programmatic management of this condition. While the study offers valuable insights for the community, several technical aspects of the manuscript, particularly within the methodology and discussion sections, require further attention from the authors.

Reviewer #3: Overall, this is an important topic that is worth reporting. However, there are some concerns that need to be addressed.

Specifically, more information is needed at the surgeon level. What are the PTT rates by surgeon? How many surgeries had each surgeon performed (not the cumulative by group)? When was the last time the BLTR surgeons were trained? supervised? audited? Were the BLTR surgeons trained by the same group that trained the PLTR surgeons?

It is well known that surgical training quality has changed substantially in the last decade (for the better). The data as presented suggest that the BLTR surgeons were trained nearly a decade ago while all of the PLTR surgeons were trained within the last year. Thus, it is somewhat of an unfair comparison to evaluate PLTR surgeons who were trained less than a year ago with state of the art methods versus surgeons who were trained under the old training system, particularly in an area where historically training was not done well and surgeons had higher rates of PTT than those trained elsewhere or by different organizations. Did all of the surgeons know both procedures?

Additionally, were all surgeons wearing loupes? If the BLTR surgeons are older and are not using loupes, this can have a substantial impact on surgical quality. If PLTR already had an edge at 4 weeks, this suggests poor surgical quality in the BLTR group, not a poor surgical procedure. Further, the imbalance in the number of surgeons between the 2 groups creates a problem - The PLTR group is much larger, which will help to balance out any outliers, while the BLTR group is smaller. Provision of surgeon specific data as well as discussion of the significant limitations of conducting an analysis with surgeons trained in different eras is important.

A few specific comments are as follows:

Abstract: it is recommended that the abstract be revised based on suggestions provided for the main text. Other abstract suggestions include: change "TT control program" to TT surgery program or Trachoma Control Program; provide details on the adverse events reported - individually.

Author summary: please review this for technical accuracy. At present, the description is not scientifically correct ("scarring of the covering of the white part of the eye and inner surface of the eyelids"? "out-turn the eyelashes"? "eyelash scratching returns"))

lines 223-227 are a bit unclear - it is standard practice to utilize prospectively collected data in a clinical trial, so it is unclear what the rationale is for explaining that they are using prospectively collective data versus "potentially inconsistent" adverse event reporting data. (Of note, this raises a flag about the quality of data if the authors feel that adverse events were not properly recorded.)

PLOS authors have the option to publish the peer review history of their article (what does this mean? ). If published, this will include your full peer review and any attached files.

**Do you want your identity to be public for this peer review?** For information about this choice, including consent withdrawal, please see our Privacy Policy .

Reviewer #1: **Yes: ** David Mabey

Reviewer #2: **Yes: ** Ehsan Ghasemian

Reviewer #3: No

---

## [Editor Report · Decision Letter 1]

19 May 2025

Dear Dr. Kempen,

We are pleased to inform you that your manuscript 'Outcomes of Posterior Lamellar Tarsal Rotation vs Bilamellar Tarsal Rotation for Trachomatous Trichiasis' has been provisionally accepted for publication in PLOS Neglected Tropical Diseases.

Best regards,

Aleksandra Inic-Kanada

Guest Editor

Stuart Blacksell

Section Editor

Shaden Kamhawi

co-Editor-in-Chief

Paul Brindley

co-Editor-in-Chief

---

## [Editor Report · Acceptance letter]

Dear Dr. Kempen,

We are delighted to inform you that your manuscript, "Outcomes of Posterior Lamellar Tarsal Rotation vs Bilamellar Tarsal Rotation for Trachomatous Trichiasis," has been formally accepted for publication in PLOS Neglected Tropical Diseases.

Best regards,

Shaden Kamhawi

co-Editor-in-Chief

Paul Brindley

co-Editor-in-Chief
